# DPAR: Decoupled Graph Neural Networks with Node-Level Differential Privacy

## ABSTRACT

Graph Neural Networks (GNNs) have achieved great success in learning with graph-structured data. Privacy concerns have also been raised for the trained models which could expose the sensitive information of graphs including both node features and the structure information. In this paper, we aim to achieve node-level differential privacy (DP) for training GNNs so that a node and its edges are protected. Node DP is inherently difficult for GNNs because all direct and multi-hop neighbors participate in the calculation of gradients for each node via layer-wise message passing and there is no bound on how many direct and multi-hop neighbors a node can have, so existing DP methods will result in high privacy cost or poor utility due to high node sensitivity. We propose a **D**ecoupled GNN with Differentially **P**rivate **A**pproximate Personalized Page**R**ank (DPAR) for training GNNs with an enhanced privacy-utility tradeoff. The key idea is to decouple the feature projection and message passing via a DP PageRank algorithm which learns the structure information and uses the top-$K$ neighbors determined by the PageRank for feature aggregation. By capturing the most important neighbors for each node and avoiding the layer-wise message passing, it bounds the node sensitivity and achieves improved privacy-utility tradeoff compared to layer-wise perturbation based methods. We theoretically analyze the node DP guarantee for the two processes combined together and empirically demonstrate better utilities with the same levels of node DP compared with existing methods.

## CCS CONCEPTS

• **Security and privacy → Privacy protections**; **Trust frameworks**.

## KEYWORDS

Differential Privacy; Graph Neural Networks; PageRank

**ACM Reference Format:**
Anonymous Author(s). 2023. DPAR: Decoupled Graph Neural Networks with Node-Level Differential Privacy. In *Proceedings of ACM Conference (Conference'17)*. ACM, New York, NY, USA, 12 pages. https://doi.org/10.1145/nnnnnnn.nnnnnnn

## 1 INTRODUCTION

Graph Neural Networks (GNNs) have demonstrated superior performance in mining graph-structured data and learning graph representations for downstream inference tasks including node classification, link prediction, and graph classification [7, 20, 29, 37]. Similar to the privacy concerns that neural network models trained on private datasets could expose sensitive information of the training data, GNN models trained on graph data that embed both the node features and graph topology information are also subject to different types of privacy attacks [36, 44, 45].

Differential privacy (DP) has become a de facto framework for training neural networks with rigorous privacy protection for the training data [1, 14]. A widely used technique is DP stochastic gradient descent (DP-SGD) [1, 40] which injects calibrated noise into the gradients during SGD-based training. Standard DP ensures that there is a bounded risk for an adversary to infer from a trained model whether a record is used in training the model. For graph data, since both node features (e.g., personal attributes) and edges (e.g., social relationships) may contain sensitive information, our goal is to achieve node-level DP (node DP), so that the risk to infer whether a node and its connecting edges are used in training the model is bounded.

**Challenges.** Achieving node DP for GNNs is inherently challenging. Unlike grid-based data such as images, graph data contains both feature vectors for each node and the edges that connect the nodes. During the training of GNN models, all direct and multi-hop neighbors participate in the calculation of gradients for each node via recursive layer-wise message passing [20, 37]. At each layer, each node aggregates the features (or the latent representations) from its neighbors when generating its own representation. There is no bound on how many direct and multi-hop neighbors a node can have. This means the sensitivity of the gradient due to the presence or absence of a node can be extremely high due to the node itself and its neighbors (or correlations between the nodes), which makes standard DP-SGD based methods [1, 40] infeasible, resulting in either high privacy cost or poor utility due to the large required DP noise.

Few recent works tackled node DP for training GNNs and they mainly attempted to bound the correlations during training to help bound the sensitivity or privacy cost. Daigavane et al. [10] sample subgraphs to ensure that each node has a bounded number of neighbors within each subgraph and limit the occurrences of each node in other subgraphs by extending the privacy-by-amplification technique [4, 23] to GNN. Their method is limited to GNNs with only one or two layers. The GAP algorithm [34] assumes a maximum degree for each node in order to bound the sensitivity of individual nodes. Meanwhile, their message-passing scheme requires DP noise at each step, therefore, it further bounds the sensitivity by bounding the number of hops. This affects the model utility as it may restrict each node from acquiring useful information from

higher hop neighbors. In sum, these approaches make it feasible to train GNNs with node DP but still sacrifice the model accuracy due to the restrictions on the number of hops during training.

**Contributions.** We propose a Decoupled GNN with Differentially Private Approximate Personalized PageRank (DPAR) for training GNNs with node DP and enhanced privacy-utility tradeoff. The key idea is to decouple the feature aggregation and message passing into two processes: 1) use a DP Approximate Personalized PageRank (APPR) algorithm to learn the structure information, and 2) use the top-$K$ neighbors determined by the APPR for feature aggregation and model learning with DP. In other words, the APPR learns the influence score of all direct and multi-hop neighbors, and the layer-wise message-passing is replaced by neighborhood aggregation based on the APPR.

Our framework is based on the decoupled GNN training framework [7, 25] which are originally designed to scale up the training for large graphs. Our main insight is that this decoupled strategy can be exploited to improve the design of DP algorithms. By capturing the most important neighbors for each node (bounds the node sensitivity) and avoiding the expensive privacy cost accumulation from the layer-wise message passing, our framework achieves enhanced privacy-utility tradeoff compared to layer-wise perturbation based methods.

Adding DP to this decoupled framework is nontrivial and presents several challenges. First, there are no existing works for computing sparsified APPR with formal node DP. While there exist DP top-$K$ selection algorithms [13], directly applying it can result in poor accuracy due to the high sensitivity since each node (and its edges) can affect all the elements in the APPR matrix. Second, while DP-SGD can be used for feature aggregation, the neighborhood sampling returns a correlated batch of nodes based on the APPR, making the privacy analysis more complex, particularly for quantifying the privacy amplification ratio. To address these challenges, we develop DP-APPR algorithms to compute the top-$K$ sparsified APPR with DP. We then utilize DP-SGD [1] for feature aggregation and model training to protect node features. We analyze the privacy loss caused by the neighborhood sampling and calibrate tighter Gaussian noise for the clipped gradients to provide a rigorous overall privacy guarantee. We summarize our contributions as follows.

- We propose DPAR, a novel de-coupled DP framework with sparsification for training GNNs with rigorous node DP. DPAR decouples message passing from feature aggregation via DP APPR and uses the top-$K$ neighbors determined by APPR for feature aggregation, which captures the most important neighbors for each node and avoids the layer-wise message passing and achieves better privacy-utility tradeoff than existing layer-wise perturbation based methods.
- We develop two DP APPR algorithms based on the exponential mechanism and Gaussian mechanism, for selecting top-$K$ elements in the APPR vector with formal node DP. We employ sampling and clipping to address the high sensitivity challenge. We utilize the exponential mechanism [13, 14] to select the indices of the top-$K$ elements first, and then compute the corresponding noisy values with additional privacy costs. The Gaussian mechanism directly adds noise to the APPR vector and then selects the

top-$K$ from the noisy vectors. We formally analyze the privacy guarantee for both methods.
- We use DP-SGD for feature aggregation and model learning based on the DP APPR. By using sparsified DP APPR vectors, we limit the maximum number of nodes one node can affect during gradient computation, which is the maximum column-wise $l_0$ norm of the DP APPR matrix. We incorporate additional clipping to ensure a maximum $\ell_1$ norm per column which determines the sensitivity of each node. We calibrate the Gaussian noise by theoretically analyzing the privacy loss and privacy amplification caused by the neighborhood sampling determined by the DP APPR and provide a rigorous overall privacy guarantee for DPAR.
- We conduct extensive experiments on five real-world graph datasets to evaluate the effectiveness of the proposed algorithms. Results show that they achieve better accuracy at the same level of node DP compared to the state-of-the-art algorithms. We also illustrate the privacy protection of the trained models.

## 2 BACKGROUND

### 2.1 GNNs with Personalized PageRank

Given a graph $G = (V, E, X)$, where V and E denote the set of vertices and edges, respectively, and $X \in \mathbb{R}^{|V| \times d}$ represents the feature matrix where each row corresponds to the associated feature vector $X_v \in \mathbb{R}^d$ ($v = 1, \ldots, |V|$) of node $v$. Each node is associated with a class (label) vector $Y_v \in \mathbb{R}^c$, such as the one-hot encoding vector, with the number of classes c. Considering the node classification task as an instance, a GNN model learns a representation function $f$ that generates the node embedding $h_v$ for each node $v \in V$ based on the features of itself as well as all its neighbors [37], and the generated node embeddings will further be used to label the class of unlabeled nodes using the softmax classifier with the cross-entropy loss.

GNN models use the recursive message-passing procedure to spread information through a graph, which couples the neighborhood aggregation and feature transformation for node representation learning. This coupling pattern can cause some potential issues in model training, including neighbor explosion and over-smoothing [7, 29]. Recent works propose to decouple the neighborhood aggregation process from feature transformation and achieve superior performance [7, 12]. Bojchevski et al. [7] show that neighborhood aggregation/propagation based on personalized PageRank [19] can maintain the influence score of all "neighboring" (relevant) nodes that are reachable to the source node in the graph, without the explicit message-passing procedure. They pre-compute a pagerank matrix $\Pi$ and truncate it by keeping only the top $k$ largest entries of each row and setting others to zero to get a sparse matrix $\Pi^{ppr}$, which is then used to aggregate node representations, generated using a neural network, of "neighbors" (most relevant nodes) to get final predictions, expressed as follows:

$$z_v = \text{softmax}\left(\sum_{u \in \mathcal{N}^k(v)} \pi'(v)_u H_{u,:}\right), \tag{1}$$

where $\mathcal{N}^k(v)$ enumerates indices of the $k$ non-zero entries in $\pi'(v)$ which is the $v$-th row of $\Pi^{ppr}$ corresponding to the node $v$'s sparse APPR vector. $H_{u,:}$ is the node representation generated by a

neural network $f_\theta$ using the node feature vector $X_u$ of each node $u$ independently.

## 2.2 Differential Privacy (DP)

DP [9, 14] has demonstrated itself as a strong and rigorous privacy framework for aggregate data analysis in many applications. DP ensures the output distributions of an algorithm are indistinguishable with a certain probability when the input datasets are differing in only one record.

DEFINITION 1. *((ϵ, δ)-Differential Privacy) [14]. Let $\mathcal{D}$ and $\mathcal{D}'$ be two neighboring datasets that differ in at most one entry. A randomized algorithm $\mathcal{A}$ satisfies (ϵ, δ)-differential privacy if for all $\mathcal{S} \subseteq Range(\mathcal{A})$:*

$$Pr\left[\mathcal{A}(\mathcal{D}) \in \mathcal{S}\right] \leq e^\epsilon Pr\left[\mathcal{A}(\mathcal{D}') \in \mathcal{S}\right] + \delta,$$

*where $\mathcal{A}(\mathcal{D})$ represents the output of $\mathcal{A}$ with the input $\mathcal{D}$, ϵ and δ are the privacy parameters (or privacy budget) and a lower ϵ and δ indicate stronger privacy and lower privacy loss.*

In this paper, we aim to achieve node-level DP for graph data to protect both the features and edges of a node.

DEFINITION 2. *((ϵ, δ)-Node-level Differential Privacy) Let $\mathcal{G}$ and $\mathcal{G}'$ be two neighboring graphs that differ in at most one node including its feature vector and all its connected edges. A randomized algorithm $\mathcal{A}$ satisfies (ϵ, δ)-node-level DP if for all $\mathcal{S} \subseteq Range(\mathcal{A})$:*

$$Pr\left[\mathcal{A}(\mathcal{G}) \in \mathcal{S}\right] \leq e^\epsilon Pr\left[\mathcal{A}(\mathcal{G}') \in \mathcal{S}\right] + \delta,$$

*where $\mathcal{A}(\mathcal{G})$ represents the output of $\mathcal{A}$ with the input graph $\mathcal{G}$.*

## 2.3 DP-SGD and Challenges

A widely used technique for achieving DP for deep learning models is DP stochastic gradient descent (DP-SGD) algorithm [1, 26]. It first computes the gradient $\mathbf{g}(x_i)$ for each example $x_i$ in the randomly sampled batch with size $B$, and then clips the $\ell_2$ norm of each gradient with a clipping threshold $C$ to bound the sensitivity of $\mathbf{g}(x_i)$ to $C$. The clipped gradient $\overline{\mathbf{g}}(x_i)$ of each example will be summed together and added with the Gaussian noise $\mathcal{N}(0, \sigma^2 C^2 \mathbf{I})$ to protect privacy. Finally, the average of the noisy accumulated gradient $\tilde{\mathbf{g}}$ will be used to update the model parameters for this step. We express $\tilde{\mathbf{g}}$ as:

$$\tilde{\mathbf{g}} \leftarrow \frac{1}{B}\left(\sum_{i=1}^{B} \overline{\mathbf{g}}(x_i) + \mathcal{N}(0, \sigma^2 C^2 \mathbf{I})\right). \quad (2)$$

In DP-SGD, each example individually calculates its gradient, e.g., only the features of $x_i$ will be used to compute the gradient $\mathbf{g}(x_i)$ for $x_i$. However, when training GNNs, nodes are no longer independent, and one node's feature will affect the gradients of other nodes. In a GNN model with $K$ layers, one node has the chance to utilize additional features from all its neighbors up to $K$-hop when calculating its gradient. Rethinking Equation 2, the bound of the sensitivity of $\sum_{i=1}^{B} \overline{\mathbf{g}}(x_i)$ becomes $B*C$ since changing one node could potentially change the gradients of all nodes in the batch $\sum_{i=1}^{B} \overline{\mathbf{g}}(x_i)$. Substituting $B*C$ for $C$ in Equation 2 and we get the following equation:

$$\tilde{\mathbf{g}}' \leftarrow \frac{1}{B}\left(\sum_{i=1}^{B} \overline{\mathbf{g}}(x_i) + \mathcal{N}(0, \sigma^2 B^2 C^2 \mathbf{I})\right). \quad (3)$$

Comparing Equation 3 to 2, to achieve the same level of privacy at each step during DP-SGD, the standard deviation of the Gaussian noise added to the gradients is scaled up by a factor of the batch size $B$, resulting in poor utility. Existing works [10, 34] mitigate the high sensitivity by bounding the number of hops and node degrees but also sacrifice the information that can be learned from higher hop neighbors, resulting in limited success in improving accuracy.

## 3 DPAR

We present our DPAR approach for training DP GNN models via DP approximate personalized PageRank (APPR). The key idea is to exploit the decoupled framework (Section 2.1) and decouple message passing from feature aggregation into two steps: 1) use a DP APPR algorithm to learn the structure information (Section 3.1), and 2) use the top-$K$ neighbors determined by the APPR for feature aggregation and model learning with DP-SGD (Section 3.2). By capturing the most important neighbors for each node from the APPR and avoiding explicit message passing, it bounds the node sensitivity without sacrificing model accuracy, achieving an improved privacy-utility tradeoff. The overall privacy budget will be split between the two steps, and we theoretically analyze the node DP guarantee for the entire framework in Section 3.2.

## 3.1 Differentially Private APPR

We develop our DP APPR algorithms based on the ISTA algorithm [16] for computing APPR. Andersen et al. [3] proposed the first approximate personalized PageRank (APPR) algorithm which is adopted in [7, 25] to replace the explicit message-passing procedure for GNNs. Most recently, Fountoulakis et al. [16] demonstrated that the APPR algorithm can be characterized as an $ell_1$-regularized optimization problem, and propose an iterative shrinkage-thresholding algorithm (ISTA) (Algorithm 3 in [16]) to solve it with a running time independent of the size of the graph. The input of ISTA contains the adjacency matrix of a graph and the one-hot vector corresponding to the index of one node in the graph, and the output is the APPR vector of that node. We develop our DP APPR algorithm based on ISTA due to its status as one of the state-of-the-art APPR algorithms. ISTA provides an excellent balance between scalability and approximation guarantees. Moreover, the resulting sparse APPR matrix can be easily accommodated into the memory, facilitating the subsequent neural network training.

Recall the purpose of calculating APPR vectors is to utilize them to aggregate representations from relevant nodes for the source node during model training. The index of each entry in an APPR vector indicates the index of the same node in the graph, and the value of each entry reflects the importance or relevance of this node to the source node. By reserving the top $K$ largest entries for each APPR vector, it is equivalent to computing a weighted average of the representations of the $K$ most relevant nodes to the source node (recall Equation 1). The graph structure information is encoded in both the indexes and values of non-zero entries in each sparse APPR vector. Thus, to provide DP protection for the graph structure, we propose two DP APPR algorithms to obtain the top-$K$ indexes and values for each APPR vector.

**Exponential Mechanism (DP-APPR-EM).** We present the DP APPR algorithm using the exponential mechanism. While we can

**Algorithm 1:** DP-APPR using the Exponential Mechanism (DP-APPR-EM)

**Input:** ISTA hyperparameters: $\gamma, \alpha, \rho$; privacy parameters: $\epsilon, \epsilon_2, \delta$; clip bound $C_2$, a graph $(V, E)$ where $V = \{v_1, ..., v_N\}$, an integer $K > 0$ and an integer $M \in [1, N]$.

1 **Initialize** the APPR matrix $\Pi \in \mathbb{R}^{M \times N}$ with all zeros.

2 **for** $i = 1, ..., M$ **do**

3      **Compute APPR:**

4      Compute the APPR vector $\mathbf{p}_{(v_i)}$ for node $v_i$ using ISTA;

5      **Clip Norm:**

6      $\hat{\mathbf{p}}_{(v_i)} \leftarrow$: for each entry $\mathbf{p}_{(v_i)}[j], j \in [1, ..., N]$, in $\mathbf{p}_{(v_i)}$, set $\mathbf{p}_{(v_i)}[j] = \mathbf{p}_{(v_i)}[j] / \max\left(1, \frac{\left\|\mathbf{p}_{(v_i)}[j]\right\|_1}{C_2}\right)$

7      **Add Noise:**

8      $\tilde{\mathbf{p}}_{(v_i)} \leftarrow \hat{\mathbf{p}}_{(v_i)} + Gumbel(\beta \mathbf{I})$, where $\beta = C_2/\epsilon$;

9      **Report Noisy Indexes:**

10      $\mathbf{N}_K \leftarrow$: select the indexes of the top $K$ entries with the largest values in $\tilde{\mathbf{p}}_{(v_i)}$;

11      **Report Noisy Values:**

12      option I: $\tilde{\mathbf{p}}'_{(v_i)} \leftarrow$: set $\hat{\mathbf{p}}_{(v_i)}[j], j \in \mathbf{N}_K$, to be $1/K$, and other entries to be 0;

13      option II: $\tilde{\mathbf{p}}'_{(v_i)} \leftarrow$: set $\hat{\mathbf{p}}_{(v_i)}[j], j \in \mathbf{N}_K$, to be $\hat{\mathbf{p}}_{(v_i)}[j] + Laplace(KC_2/\epsilon_2)$, and other entries to be 0;

14      **Replace** the $i$-th row of $\Pi$ with $\tilde{\mathbf{p}}'_{(v_i)}$.

15 **end**

16 **return** $\Pi$ and the overall privacy cost.

employ a DP top-$K$ selection algorithm based on the exponential mechanism [13], there are several challenges that need to be addressed. First, each node (and its edges) can change an arbitrary number of elements in the APPR vector and lead to significant changes in each element. Second, each node can change an arbitrary number of APPR vectors in the APPR matrix. Both of these mean extremely high sensitivity, making a direct application of the top-$K$ selection algorithm ineffective. To address them, we employ two techniques: 1) clipping each element to bound the sensitivity, 2) sampling and only computing APPR for a subset of M nodes in the graph to reduce sensitivity. We then employ the exponential mechanism to select the top=$K$ values.

As shown in Algorithm 1, for each of the M sampled nodes, we first compute the APPR vector using the ISTA algorithm (line 4). Then we employ clipping to bound the sensitivity of each element by C2 (line 6). We use the clipped value as its utility score for the exponential mechanism since the magnitude of each entry indicates its importance (utility) and is used as the weight when aggregating the representation of the nodes. We simulate the exponential mechanism by injecting a one-shot Gumbel noise to the clipped vector $\hat{\mathbf{p}}_{(v)}$ (line 8) and then select the indexes of top $K$ largest noisy entries [13] (line 10). We can then either: option I) set the values of all top $K$ entries to be $1/K$, which means we consider the top $K$ entries equally important to the source node, or option II) spend additional privacy budget to obtain the noisy values of the top $K$ entries with DP. Given the same total privacy budget, the option I has a better chance to output indexes of the actual top $K$ entries while losing the importance scores. In contrast, option II

sacrifices some accuracy in selecting the indexes of top $K$ entries but has additional importance scores.

**Privacy Analysis of DP-APPR-EM.** We formally analyze the DP guarantee of Algorithm 1 utilizing the following corollary for the exponential mechanism based top-$K$ selection.

COROLLARY 1. *[13]* $\mathcal{M}^k_{Gumbel}(u)$ *adds the one-shot* $Gumbel(\Delta(u)/\epsilon)$ *noise to each utility score* $u(x, r)$ *and outputs the k indices with the largest noisy values. For any* $\delta \geq 0$, $\mathcal{M}^k_{Gumbel}(u)$ *is* $(\epsilon', \delta)$-*DP where*

$$\epsilon' = 2 \cdot \min\left\{k\epsilon, k\epsilon\left(\frac{e^{2\epsilon}-1}{e^{2\epsilon}+1}\right) + \epsilon\sqrt{2k\ln(1/\delta)}\right\}$$

The privacy analysis conducted in [13] assumes independent users and the sensitivity $\Delta(u)$ is 1. In our case, each node (and its edges) can modify an arbitrary number of elements in the APPR vector and each element can change at most by $C_2$ due to clipping (line 6). Consequently, the sensitivity $\Delta(u)$ used in Corollary 1 is set to $C_2$ and the noise is calibrated accordingly in our algorithm (line 8). Additionally, since each node can change up to M vectors in the APPR matrix, we use sequential composition to bound the privacy loss for M APPR vectors. With the calibrated noise and composition, we establish the DP guarantee in Theorem 1.

THEOREM 1. *For any* $\epsilon > 0$, $\epsilon_2 > 0$ *and* $\delta \in (0, 1]$, *let* $\epsilon_1 = 2 \cdot \min\left\{K\epsilon, K\epsilon\left(\frac{e^{2\epsilon}-1}{e^{2\epsilon}+1}\right) + \epsilon\sqrt{2K\ln(1/\delta)}\right\}$, *Algorithm 1 is* $(\epsilon_{g_1}, 2M\delta)$-*differentially private for option I, and* $(\epsilon_{g_2}, 2M\delta)$-*differentially private for option II, where* $\epsilon_1 = \epsilon_{g_1} / \left(2\sqrt{M\ln\left(e + \epsilon_{g_1}/2M\delta\right)}\right)$ *and* $\epsilon_1 + \epsilon_2 = \epsilon_{g_2} / \left(2\sqrt{M\ln\left(e + \epsilon_{g_2}/2M\delta\right)}\right)$.

PROOF. See Appendix 7.1 for the proof. □

**Gaussian Mechanism.** We explore another DP-APPR algorithm (DP-APPR-GM) based on the Gaussian mechanism [14] and output perturbation. The idea behind DP-APPR-GM is to use the clipping strategy to bound the global sensitivity of each output PageRank vector and add Gaussian noise to each bounded PageRank vector to achieve DP. See Appendix 7.2 for more details about DP-APPR-GM. We omit them here due to space limitations.

## 3.2 Differentially Private GNNs

We show our overall approach for training a DP GNN model in Algorithm 2. The main idea is to use DP APPR for neighborhood sampling and then use DP-SGD to achieve DP for the node features. We employ additional sampling and clipping to reduce the privacy cost.

Given the graph dataset $\overline{G}$, we first use a sampling rate $q'$ to randomly sample nodes from $\overline{G}$ to form a subgraph $G = (V, E, X)$ containing only the sampled nodes and their connected edges, which is used for training in Algorithm 3. This sampling step brings a privacy amplification effect in our privacy guarantee by multiplying a factor of $q'$ [5, 23]. Note that this is different from the batch sampling during each iteration of the training process. We further sample M nodes to compute the DP APPR using DP-APPR-EM or DP-APPR-GM and use it as input for Algorithm 2.

Utilizing the sparsified DP APPR vectors (each row has only top $K$ non-zero elements) limits the impact of a node on the gradient

---

**Algorithm 2:** Differentially Private GNNs

---

**Input:** The graph dataset $\overline{G}$, sampling rate $q'$, randomly sampled training graph $G = (V, E, X)$ from $\overline{G}$ by $q'$ where $V = \{v_1, ..., v_N\}$, a sampled subset $V_M \subseteq V$ with size $M$ (for computing APPR), learning rate $\eta_t$, batch size $B$, training steps $T$, noise scale $\sigma$, gradient norm bound $C$, clip bound $\tau$, the DP APPR matrix $\Pi \in \mathbb{R}^{M \times N}$ of $V_M$ satisfying $(\epsilon_{pr}, \delta_{pr})$-DP.

1 **Initialize** $\theta_0$ randomly
2 **for** $j = 1, ..., N$ **do**
3      $\Pi_{:,j} \leftarrow \Pi_{:,j} / \max\left(1, \frac{\|\Pi_{:,j}\|_1}{\tau}\right)$
4 **end**
5 **for** $t = 1, ..., T$ **do**
6      Take a randomly sampled batch $B$ and their $K$ neighbors based on $\Pi$ from $V_M$.
7      **Compute Gradient:**
8      For each $i \in B_t$, compute $\mathbf{g}_t(v_i) \leftarrow \nabla_{\theta_t} \mathcal{L}(\theta_t, v_i)$.
9      **Clip Gradient:**
10      $\overline{\mathbf{g}}_t(v_i) \leftarrow \mathbf{g}_t(v_i) / \max\left(1, \frac{\|\mathbf{g}_t(v_i)\|_2}{C}\right)$.
11      **Add Noise:**
12      $\tilde{\mathbf{g}}_t \leftarrow \frac{1}{B}\left(\sum_i \overline{\mathbf{g}}_t(v_i) + \mathcal{N}(0, \sigma^2 C^2 \mathbf{I})\right)$.
13      **Update Parameters:**
14      $\theta_{t+1} \leftarrow \theta_t - \eta_t \tilde{\mathbf{g}}_t$.
15 **end**
16 **return** $\theta_T$ and the overall privacy cost.

---

computation of up to $B'$ nodes, where $B'$ is the maximum column-wise $l_0$ norm of the DP APPR matrix (number of non-zero elements in each column). The exact impact or sensitivity is determined by the maximum column-wise $\ell_1$ norm of the DP APPR matrix (see privacy analysis for more details). Hence, we employ additional clipping on the DP APPR matrix to bound the sensitivity. Given $\Pi$ computed using DP-APPR algorithms, each column of $\Pi$ is clipped to have a maximum $\ell_1$ norm of $\tau$ to limit privacy loss (line 4).

During each step of the training, we randomly sample a batch of $B$ nodes and their neighbors (both direct and indirect) based on the APPR vectors, and the features of up to $B * K$ nodes are loaded into memory for gradient computation (line 6). The loss function $\mathcal{L}(\theta, v_i)$ is the cross-entropy between node $v_i$'s true label and its prediction from Equation 1. Following DP-SGD, the gradient for each node in the batch is computed, clipped to have a maximum $\ell_2$ norm of $C$, and added with Gaussian noise of sensitivity $C$ (line 7-12). The model is then updated with the average noisy gradient (line 14).

**Privacy Analysis.** Theorem 2 presents the DP analysis of Algorithm 2. An essential distinction between our algorithm and the original DP-SGD is that our neighborhood sampling returns a correlated batch of nodes for gradient computation (i.e., the computation of $\mathbf{g}_t(v_i)$ requires the features of the neighboring nodes of node $v_i$, and node $v_i$ accesses the fixed $K$ nodes based on the DP-APPR vector), while the original DP-SGD uses the much simpler Poisson sampling. As a result, the privacy analysis of our algorithm is more involved, especially in terms of quantifying the privacy amplification ratio under such a neighbor-correlated sampling setting. We

prove that the privacy amplification ratio is proportional to the maximum of the column-wise $\ell_1$ norm of the DP-APPR matrix.

For the composition of DP-APPR and DP-SGD, we use the standard composition theorem. Recall that for the privacy composition of multiple DP-APPR vectors for the privacy of DP-APPR (Theorem 1 and 2), we used a strong composition theorem. We note that our privacy analysis can always benefit from a more advanced composition theorem to achieve tighter overall privacy, which can be a future work direction.

**THEOREM 2.** *There exist constants $c_1$ and $c_2$ so that given probability $q = B/N$ and the number of steps $T$, for any $\epsilon_{sgd} < c_1 q^2 T$, Algorithm 2 is $q'(\epsilon_{sgd} + \epsilon_{pr}, \delta_{sgd} + \delta_{pr})$ -differentially private corresponding to $\overline{G}$, for any $\delta_{sgd} > 0$ if we choose $\sigma \geq c_2 \frac{q\tau\sqrt{T \log(1/\delta_{sgd})}}{\epsilon_{sgd}}$.*

PROOF. See Appendix 7.3 for the proof. □

## 4 EXPERIMENTAL RESULTS

We evaluate our method on five graph datasets with varying sizes and edge density: Cora-ML [6], Microsoft Academic graph [35], CS, [35], Physics [35] and Reddit [20]. Appendix 7.4 provides the details of each dataset.

**Setup.** To simulate the real-world situations where training nodes are assumed to be private and not publicly available, we split the nodes into a training set (80%) and a test set (20%), and select inductive graph learning setting by removing edges between the two sets. The training nodes are inaccessible during inference. We use the same 2-layer feed-forward neural network with a hidden layer size of 32 as in [7] for all datasets. The training epochs are fixed at 200, the learning rate at 0.005, and the batch size at 60. The hyperparameters for ISTA are chosen through grid search as $\alpha = 0.25$, $\rho = 10^{-4}$, and $\gamma = 10^{-4}$. In our comparison with baseline methods, we set $K$ to 2 for computing top-$K$ sparsified DP APPR. We also present results on the effect of $K$ with different $K$ values. The graph sampling rate is set to $q' = 9\%$ for all datasets, and $M = 70$ nodes are chosen randomly and uniformly to generate DP-APPR vectors. Experiments are conducted on a server with an Nvidia K80 GPU, a 6-core Intel CPU, and 56 GiB RAM. Results are based on the mean of 10 independent trials. The source code is available[1].

**Our Approach and Baselines.** Our proposed algorithms using the DP-APPR with exponential mechanism (options I and II in Algorithm 1) are referred to as **DPAR-EM0** and **DPAR-EM1**, respectively, and our algorithm using the DP-APPR with Gaussian mechanism is referred to as **DPAR-GM**.

We compare our proposed algorithms with two state-of-the-art methods achieving node DP for GNN and one baseline method: 1) **SAGE** [10] samples subgraphs of 1-hop neighbors of each node to train 1-layer GNNs with the GraphSAGE [20] model. 2) **GAP** [34] uses aggregation perturbation and MLP-based encoder and classifier with DP-SGD and a bounded node degree and the number of hops. 3) **Features** is a baseline method that only uses node feature as an independent input to train the GNN model and does not consider the structural information of the graph. It utilizes the original DP-SGD to achieve node DP. Note that **Features** is equal to the case where we use the one-hot label indicator vector as each node's

---

[1]The source code is available at: https://anonymous.4open.science/r/anonymous-D626.

**Table 1: Privacy budget and test accuracy on each graph dataset**

| Dataset | Privacy Budget | GAP | SAGE | Features | DPAR-EM0 | DPAR-EM1 | DPAR-GM | DPARNoDP | GAPNoDP |
|---------|----------------|-----|------|----------|----------|----------|---------|----------|---------|
| Cora-ML | $(1, 2 \times 10^{-3})$ | 0.34 | 0.152 | **0.5733** | 0.3421 | 0.2895 | 0.3333 | 0.7076 | 0.8883 |
|  | $(8, 2 \times 10^{-3})$ | 0.5733 | 0.368 | 0.6107 | 0.5965 | **0.6199** | 0.4854 |  |  |
| MS Academic | $(1, 8 \times 10^{-4})$ | 0.6563 | 0.013 | 0.83 | 0.8306 | **0.8569** | 0.8225 | 0.955 | 0.9571 |
|  | $(8, 8 \times 10^{-4})$ | 0.8581 | 0.063 | 0.8723 | 0.9054 | 0.9135 | **0.9165** |  |  |
| CS | $(1, 8 \times 10^{-4})$ | 0.66 | 0.0917 | 0.8344 | 0.8898 | 0.8921 | **0.8927** | 0.9707 | 0.9571 |
|  | $(8, 8 \times 10^{-4})$ | 0.8537 | 0.7366 | 0.895 | 0.9017 | 0.8994 | **0.9063** |  |  |
| Reddit | $(1, 1 \times 10^{-4})$ | 0.7047 | 0.086 | 0.7436 | 0.9167 | 0.9286 | **0.934** | 0.9698 | 0.9949 |
|  | $(8, 1 \times 10^{-4})$ | 0.9161 | 0.82 | 0.777 | 0.9375 | **0.9399** | 0.931 |  |  |
| Physics | $(1, 1 \times 10^{-4})$ | 0.8192 | 0.1263 | 0.8412 | 0.8887 | 0.8927 | **0.8948** | 0.9548 | 0.9597 |
|  | $(8, 1 \times 10^{-4})$ | 0.9088 | 0.8919 | 0.9017 | 0.9023 | 0.9020 | **0.9101** |  |  |

APPR vector in Algorithm 2 (i.e., no correlation with other nodes is used). We included this baseline to help characterize the datasets and calibrate the results, i.e., a good performance of the method may suggest that the topological structure of the particular dataset has limited benefit in training GNN. The models **DPARNoDP** and **GAPNoDP** indicate the versions of the respective methods (**DPAR**, **GAP**) with no DP protection.

**Inference Phase.** As suggested in [7], instead of computing the APPR vectors for all testing nodes and generating predictions based on their APPR vectors, we use power iteration during inference, e.g.,

$$Q^{(0)} = H, \quad Q^{(p)} = (1 - \alpha)D^{-1}AQ^{(p-1)} + \alpha H, p \in [1, ..., P]. \quad (4)$$

In Equation 4, $H$ is the representation matrix of testing nodes generated by the trained private model, with the input being the feature matrix of the testing nodes. $D$ and $A$ are the degree matrix and adjacency matrix of the graph containing only testing nodes, respectively. The final output of power iteration $Q^{(P)}$ will be input into a softmax layer to generate the predictions for testing nodes. We set $P = 2$ and the teleportation constant $\alpha = 0.25$ as suggested in [7] in our experiments.

## 4.1 Privacy vs. Accuracy Trade-off

We use the value of privacy budget $\epsilon$ (with fixed $\delta$ chosen to be roughly equal to the inverse of each dataset's number of training nodes) to represent the level of privacy protection and use the test accuracy for node classification to indicate the model's utility. Table 1 shows results between our proposed methods and baselines in all datasets.

From Table 1, in comparison to **GAP** and **SAGE**, our methods show superior test accuracy under the same privacy budget on all datasets. For instance, when $\epsilon = 1$, our methods (**DPAR-GM**, **DPAR-EM0**, and **DPAR-EM1**) achieve the highest test accuracy of **0.3421/0.8569/0.8927/0.934/0.8948** on Cora-ML/MS Academic/CS/Reddit/Physics datasets respectively. The best accuracy achieved by the baselines (**GAP** and **SAGE**) is 0.34/0.6563/0.66/0.7047 /0.8192 on the corresponding datasets, indicating the test accuracy improvement by **0.62%/30.6%/35.3%/32.5%/9.23%** respectively on these datasets under the same privacy protection budget. The performance improvement demonstrates our method's superior ability

to balance the privacy-utility trade-off on training graph datasets with privacy considerations.

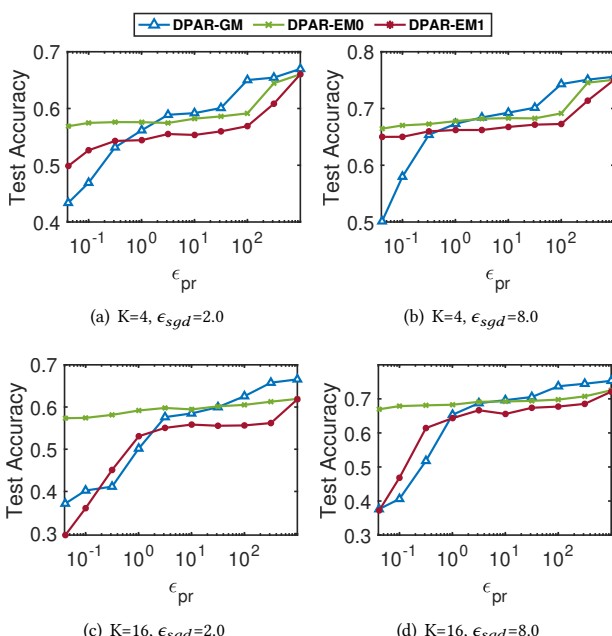

(a) K=4, $\epsilon_{sgd}$=2.0

(b) K=4, $\epsilon_{sgd}$=8.0

(c) K=16, $\epsilon_{sgd}$=2.0

(d) K=16, $\epsilon_{sgd}$=8.0

**Figure 1: Relationship between privacy budget $\epsilon$ (fixed $\delta = 2 \times 10^{-3}$) and test accuracy on Cora-ML dataset.**

Existing research in the graph neural network community suggests that features alone, especially for heterophilic graphs, can sometimes result in better-trained node classification models with MLP as the backend architecture compared to state-of-the-art GNN models [31]. For the Cora-ML dataset, which has a low edge density, the **Features** approach outperforms our methods when $\epsilon$ is small (e.g., 1). This is because our methods allocate part of the privacy budget to protect graph structure information, which may not be as critical, while **Features** uses its entire privacy budget to protect node features without considering graph structure information. However, as $\epsilon$ increases (e.g., 8), our methods outperform **Features**.

Our proposed methods protect the graph structure and node features independently via the decoupled framework. Different graphs possess unique characteristics, and the relative significance of structure information and node features can differ among them.

Accordingly, our methods are able to allocate the total privacy budget differently to protect node features and structures, which leads to more precise and tunable privacy protection for graph data that includes both feature and structural information.

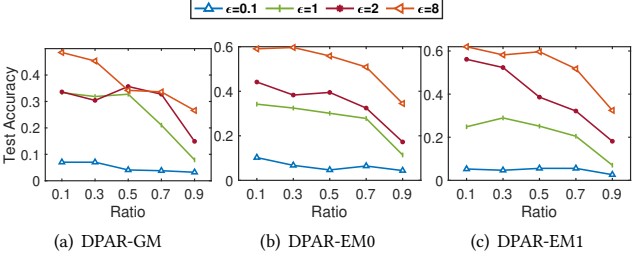

Figure 2: Cora-ML. The privacy budget $\epsilon$ ratio for DP-ARRP

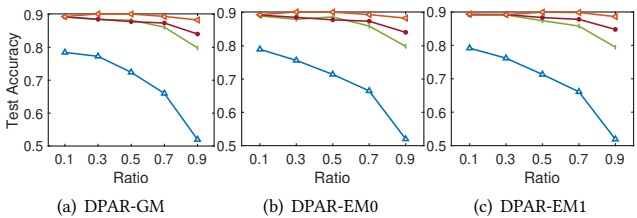

Figure 3: CS. The privacy budget $\epsilon$ ratio for DP-ARRP

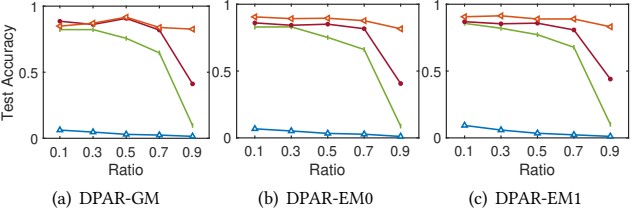

Figure 4: MS Academic. The privacy budget $\epsilon$ ratio for DP-ARRP

**Ablation Study of Different DP-APPR Methods.** To further study the impact of DP-APPR on the model accuracy, in Figure 1, we fix $\epsilon_{sgd}$ (privacy budget for DP-SGD) and use varying $\epsilon_{pr}$ (privacy budget for DP-APPR) as the x-axis. For **DPAR-GM** and **DPAR-EM1**, the higher the $\epsilon_{pr}$, the less noise is added when calculating the APPR vector for each training node. This allows a better chance for each node to aggregate representations from more important nodes using more precise importance scores. Hence these models have higher test accuracy compared to **DPAR-EM0**. While for **DPAR-EM0**, noise in DP-APPR will only affect the output of the indexes of the top $K$ most relevant nodes corresponding to the source node, but not their importance scores. **DPAR-EM0** achieves better performance than **DPAR-GM** and **DPAR-EM1** when the privacy budget $\epsilon_{pr}$ is small, this is because **DPAR-EM0** uses $1/K$ as the importance score for all nodes (considering nodes equally important), which diminishes the negative effect of less important or irrelevant nodes having high importance scores due to the noise in **DPAR-GM** and **DPAR-EM1**. Both **DPAR-EM0** and **DPAR-EM1** are based on the exponential mechanism designed for identifying the index of the top-$K$ accurately. Therefore, when the privacy budget is small, they outperform **DPAR-GM**. However, when the

privacy budget is large, they all have a good chance to find the indexes of the actual top $K$, and **DPAR-GM** becomes gradually better than **DPAR-EM0** and **DPAR-EM1**, as the Gaussian noise has better privacy loss composition property.

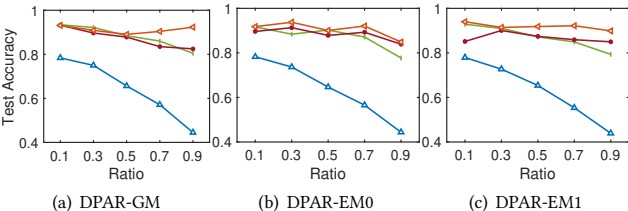

Figure 5: Reddit. The privacy budget $\epsilon$ ratio for DP-ARRP

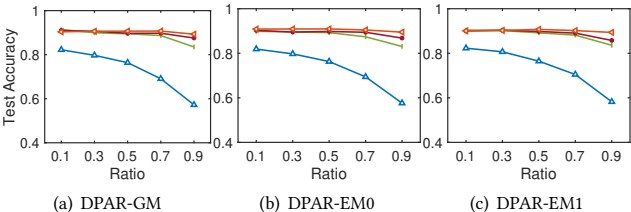

Figure 6: Physics. The privacy budget $\epsilon$ ratio for DP-ARRP

## 4.2 Privacy Protection Effectiveness

**Privacy Budget Allocation between DP-APPR and DP-SGD.** The total privacy budget is divided between DP-APPR and DP-SGD. We compare the impact of the budget allocation by changing the ratio of the total privacy budget used by each of them. Figure 2, 3, 4, 5, and 6 report the model test accuracy with varying ratios of the total privacy budget used for DP-APPR for the five datasets respectively, and they share the same legend as in Figure 2. A lower ratio means a smaller privacy budget is allocated for DP-APPR while more is allocated for DP-SGD. The impact of the ratio on the privacy-utility trade-off is closely aligned with the characteristics of each dataset. From Figure 2, the model achieves better accuracy when the ratio is lower, regardless of the total privacy budget. This is because of the characteristics of the Cora-ML dataset, as its node features are more important than its structure. Interestingly, when the privacy budget is small, Figure 3,4, 5, and 6 show that information from node features is crucial for all datasets. Allocating more privacy budget to DP-SGD can learn more useful information from the node features and improve model accuracy. When the privacy budget is large, e.g., $\epsilon = 8$, we find that in MS Adacemic and CS datasets, the model can achieve the best results when the budget is equally divided, suggesting the importance of learning from both the structure information and features.

## 4.3 Effects of Privacy Parameters

We use the Cora-ML dataset as an example to demonstrate the effects of the parameters specific to privacy, including the clipping bound in DP-APPR, the number of nodes M in DP-APPR, the number of selected top-$K$ entries in DP-APPR, the batch size in DP-SGD, and the clipping bound in DP-SGD. By default, we set the batch size to 60, the clipping bound $C_1$ in DP-APPR-GM to 0.01, the clipping bound $C_2$ in DP-APPR-EM to 0.001, the gradient norm clipping bound $C$

for DP-SGD to 1, and M to 70. We analyze them individually while keeping the rest constant as the default values.

**Clipping Bound in DP-APPR ($C_1$ and $C_2$).** Figure 7 shows the effect of clipping bound in DP-APPR on the model's test accuracy. Given a constant total privacy budget, the standard deviation of the noise added to the APPR vectors is proportional to the clipping bound ($C_1$ in DP-APPR-GM and $C_2$ in DP-APPR-EM). Hence, choosing a smaller clipping bound in general can avoid adding too much noise and result in better accuracy. However, too small of a clipping bound may degrade the accuracy due to the clipping error. In experiments, we set $C_1$ to be 0.01 and $C_2$ to be 0.001 for all datasets.

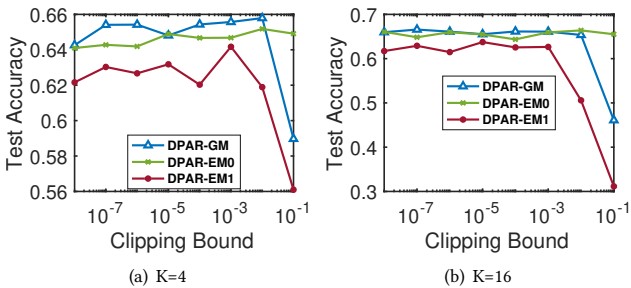

(a) K=4          (b) K=16

**Figure 7: Cora-ML. Relationship between clipping bound of DP-APPR and model test accuracy. Fix total privacy budget $(\epsilon, \delta) = (8, 2 \times 10^{-3})$.**

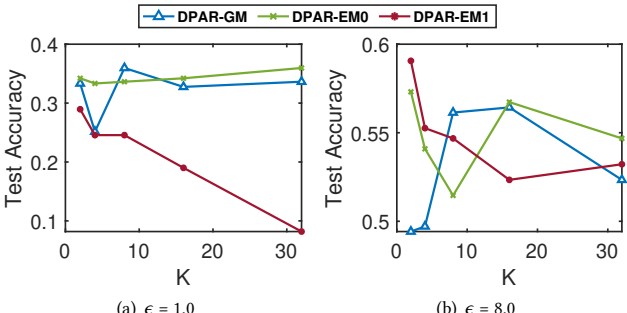

(a) $\epsilon = 1.0$          (b) $\epsilon = 8.0$

**Figure 8: Cora-ML: Relationship between the number of selected top K entries in DP-APPR vector and model test accuracy.**

**Number of Top-$K$ in DP-APPR ($K$).** Figure 8 shows the accuracy with respect to varying $K$ for the top-$K$ selection in DP-APPR. The Gaussian mechanism's sensitivity depends on the $\ell_2$ norm of the APPR vector. We use a clip bound $C_1$ to restrict the $\ell_2$ norm of the APPR vector, therefore the privacy guarantees are linked to $C_1$, not $K$. $K$ impacts the number of non-zero entries in each DP-APPR vector, influencing node feature embeddings. A small $K$ may not capture enough neighbors while a higher K may include more irrelevant nodes as "neighbors", inversely affecting aggregated information. For the Exponential mechanism, we clip each APPR vector value by $C_2$ to control sensitivity. The privacy guarantee is dependent on both $C_2$ and $K$. A larger K means more noise for each entry, affecting accuracy. From Figure 8, we can observe that DPAR-EM1 results highlight this effect, while DPAR-EM0 mitigates it by assigning a value of $1/K$ without additional noise. In our

experiments compared against baselines, we use a fixed $K = 2$ for all datasets.

We also investigate the impact of batch size in DP-SGD ($B$), the clipping bound in DP-SGD ($C$), and the number of nodes in DP-APPR ($M$). Due to space constraints, we have included the results in Appendix 7.6.

## 5 RELATED WORK

**Differentially Private Graph Publishing.** Works on privacy-preserving graph data publishing aim to release the entire graph [18, 21, 32, 38], or the statistics or properties of the original graph [2, 8, 11, 24, 30, 41], with the DP guarantee. Different from those works, our work in this paper focuses on training GNN models on private graph datasets and publishing the model that satisfies a formal node-level DP guarantee.

**Differentially Private Graph Neural Networks.** Yang et al. [39] propose to train a graph generation model using DP-SGD to generate graphs with the edge-DP guarantee that protects individual link privacy. Sajadmanesh et al. [33] develop a privacy-preserving GNN training algorithm based on local DP (LDP) to protect node features' privacy without considering node edges' privacy. Zhang et al. [43] use LDP and functional mechanism [42] to enforce privacy guarantee on user's sensitive features when training graph embedding models for the recommendation. Lin et al. [28] propose a privacy-preserving learning framework for decentralized network graphs where each local user has a local graph, to preserve LDP for every user, particularly on the notion of edge DP. Epasto et al. [15] introduce a differentially private (DP) Personalized PageRank algorithm with an edge-level DP guarantee for graph embedding. However, none of these works achieved the goal of providing strict node-level DP concerning both features and connected edges for each node in the graph when training GNN models.

Few recent works achieved node-level DP for training GNN models [10, 34] as we have discussed earlier. While they make it feasible to train GNNs with node DP, they still sacrifice the model accuracy due to the restrictions on the number of hops or layers during training. Our experiment results showed that DPAR outperforms both of these approaches.

## 6 CONCLUSION

We studied the problem of private learning for GNN models. Our method is based on a two-stage framework including DP approximate personalized PageRank and DP-SGD for protecting the graph structure information and node features respectively. We developed two DP-APPR algorithms using the Gaussian mechanism and the exponential mechanism for learning the PageRank with the most relevant neighborhood of each node. DP-APPR not only protects nodes' edge information but also limits the sensitivity of each node during the model training using DP-SGD, which facilitates DP-SGD to play its role in protecting nodes' feature information. Experimental results on real-world graph datasets demonstrate the effectiveness of our proposed methods in achieving better privacy and utility trade-off compared to state-of-the-art methods. We leave the development of better DP-APPR algorithms with tighter privacy guarantees and adaptive privacy budget allocation strategy (e.g., between DP-APPR and DP-SGD based on dataset characteristics) as future work.

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

# 7 APPENDIX

## 7.1 Proof for Theorem 1

PROOF. We first consider the privacy loss of outputting the noisy APPR vector $\tilde{\mathbf{p}}'_{(v_i)}$ for node $v_i$ in Algorithm 1. For each element in the APPR vector, we use its value as its utility score. Since each element is nonnegative and clipped by the constant $C_2$, the $ell_1$ sensitivity $\Delta(u)$ of each element is equal to $C_2$. By adding the one-shot Gumbel noise $Gumbel(\beta\mathbf{I})$ where $\beta = C_2/\epsilon$ to the clipped APPR vector $\tilde{\mathbf{p}}(v_i)$, option I selects $K$ indices with the largest noisy values and satisfies $(\epsilon_1, \delta)$-DP where $\epsilon_1 = 2 \cdot \min\left\{K\epsilon, K\epsilon\left(\frac{e^{2\epsilon}-1}{e^{2\epsilon}+1}\right) + \epsilon\sqrt{2K\ln(1/\delta)}\right\}$ according to Corollary 1. Option II uses the Laplace mechanism [14] to report $K$ selected noisy values. By adding Laplace noise $Laplace(KC_2/\epsilon_2)$ to each clipped element, option II costs an additional $\epsilon_2$ privacy budget [14] since the $ell_1$ sensitivity of each element is $C_2$, and satisfies $(\epsilon_1+\epsilon_2, \delta)$-DP.

Now we consider the privacy loss of Algorithm 1 which outputs $M$ noisy APPR vectors. We use the optimal composition theorem in [22] which argues that for $k$ sub-mechanisms, each with an $(\epsilon, \delta)$-DP guarantee, the overall privacy guarantee is $(\epsilon_g, \delta_g)$, where $\epsilon = \epsilon_g/(2\sqrt{k\ln(e + \epsilon_g/\delta_g)})$ and $\delta = \delta_g/2k$. By substituting $M$ for $k$ and $\epsilon_1$ / $\epsilon_1 + \epsilon_2$ (option I/option II) for $\epsilon$, we get the privacy loss of Algorithm 1 with option I is $(\epsilon_{g_1}, 2M\delta)$, where $\epsilon_1 = \epsilon_{g_1}/\left(2\sqrt{M\ln\left(e + \epsilon_{g_1}/2M\delta\right)}\right)$, and the privacy loss of Algorithm 1 with option II is $(\epsilon_{g_2}, 2M\delta)$, where $\epsilon_1+\epsilon_2 = \epsilon_{g_2}/\left(2\sqrt{M\ln\left(e + \epsilon_{g_2}/2M\delta\right)}\right)$. □

## 7.2 Gaussian Mechanism (DP-APPR-GM)

We propose another DP APPR algorithm (DP-APPR-GM) based on the Gaussian mechanism [14] and output perturbation. DP-APPR-GM utilizes a similar sampling and clipping strategy to limit the sensitivity of the APPR vector and directly adds Gaussian noise to each element to achieve DP. As shown in Algorithm 3, for each node $v$, we clip the $\ell_2$ norm of its APPR vector $\mathbf{p}_{(v)}$ (line 6) and add the calibrated Gaussian noise to each element in the clipped $\mathbf{p}_{(v)}$ (line 8). We then select the top $K$ largest entries in $\tilde{\mathbf{p}}_{(v)}$ to get a sparse vector $\tilde{\mathbf{p}}'_{(v)}$ (line 10).

**Privacy Analysis of DP-APPR-GM.** Using the properties of the Gaussian mechanism and the optimal composition theorem [22], we establish the overall privacy guarantee for the DP-APPR-GM algorithm. Note that the DP guarantee is independent of $K$, in contrast with DP-APPR-EM.

THEOREM 3. *Let $\epsilon > 0$ and $\delta \in (0, 1]$, Algorithm 3 is $(\epsilon_g, 2M\delta)$-differentially private where $\epsilon = \epsilon_g/\left(2\sqrt{M\ln\left(e + \epsilon_g/2M\delta\right)}\right)$.*

PROOF. We utilize the optimal composition theorem in [22] which argues that for $k$ sub-mechanisms, each with an $(\epsilon, \delta)$-DP guarantee, the overall privacy guarantee is $(\epsilon_g, \delta_g)$-DP, where $\epsilon = \epsilon_g/(2\sqrt{k\ln(e + \epsilon_g/\delta_g)})$ and $\delta = \delta_g/2k$. In Algorithm 3, the noisy APPR vector for each node satisfies $(\epsilon, \delta)$-DP by the Gaussian mechanism independently. Since the returned APPR matrix contains the noisy APPR vectors of $M$ nodes, the number of components for

---

**Algorithm 3:** DP-APPR using the Gaussian Mechanism (DP-APPR-GM)

**Input:** ISTA hyperparameters: $\gamma, \alpha, \rho$; privacy parameters: $\epsilon, \delta$; clip bound $C_1$, a graph $(V, E)$ where $V = \{v_1, ..., v_N\}$, an integer $K > 0$ and an integer $M \in [1, N]$.

1 **Initialize** the APPR matrix $\mathbf{\Pi} \in \mathbb{R}^{M \times N}$ with all zeros.
2 **for** $i = 1, ..., M$ **do**
3    **Compute APPR Vector:**
4    Compute the APPR vector $\mathbf{p}_{(v_i)}$ for node $v_i$ using ISTA;
5    **Clip Norm:**
6    $\hat{\mathbf{p}}_{(v_i)} \leftarrow \mathbf{p}_{(v_i)}/\max\left(1, \frac{\|\mathbf{p}_{(v_i)}\|_2}{C_1}\right)$;
7    **Add Noise:**
8    $\tilde{\mathbf{p}}_{(v_i)} \leftarrow \hat{\mathbf{p}}_{(v_i)} + \mathcal{N}(0, \sigma^2\mathbf{I})$, where $\sigma = \sqrt{2\ln(1.25/\delta)}C_1/\epsilon$;
9    **Sparsification:**
10    $\tilde{\mathbf{p}}'_{(v_i)} \leftarrow$: select the top $K$ largest entries in $\tilde{\mathbf{p}}_{(v_i)}$ by setting all other entries with small values to zero.
11    **Replace** the $i$-th row of $\mathbf{\Pi}$ with $\tilde{\mathbf{p}}'_{(v_i)}$.
12 **end**
13 **return** $\mathbf{\Pi}$ and compute the overall privacy cost using the optimal composition theorem.

---

composition is $M$. We substitute $M$ for k and $2M\delta$ for $\delta_g$, which can conclude the proof. □

## 7.3 Proof for Theorem 2

PROOF. Denote $\mu_0$ the Gaussian distribution with mean 0 and variance 1. Assume $\mathbb{D}'$ is the neighboring feature dataset of $\mathbb{D}$, which differs at $i^\dagger$ such that $\mathbf{x}'_{i^\dagger} \neq \mathbf{x}_{i^\dagger}$. Without loss of generality, we assume $\nabla f(\mathbf{x}_i) = \mathbf{0}$, for any $\mathbf{x}_i \in \mathbb{D}$, while $\nabla f(\mathbf{x}'_{i^\dagger}) = \mathbf{e}_1$. Recall that the DP-APPR matrix is $\mathbf{\Pi}$, where $\mathbf{\Pi}_{i:}$ is the $i$-th row and the DP-APPR vector for node $i$, while $\mathbf{\Pi}_{:j}$ is the $j$-th column of $\mathbf{\Pi}$. In addition, we can assume that $\|\mathbf{\Pi}_{:j}\|_1 \leq \tau$ due to the clipping in line 3, for all $j = 1, \ldots, n$, and denote $\mu_\tau$ the Gaussian distribution with mean $\tau$ and variance 1. Then, we have $\mathbb{E}[\mathcal{G}(\mathbb{D})]$ and $\mathbb{E}[\mathcal{G}(\mathbb{D}')]$ below,

$$
\begin{aligned}
\mathbb{E}[\mathcal{G}(\mathbb{D})] &= [\frac{|\mathcal{B}|}{n}\sum_{j \neq i^\dagger, j \notin \mathcal{N}(i^\dagger)} G_j] + [\frac{|\mathcal{B}|}{n}\sum_{j \neq i^\dagger, j \in \mathcal{N}(i^\dagger)} G_j] + [\frac{|\mathcal{B}|}{n}G_i] \\
&= [\frac{|\mathcal{B}|}{n}\sum_{j \neq i^\dagger, j \notin \mathcal{N}(i^\dagger)}\sum_{k \in \mathcal{N}(j)} \mathbf{\Pi}_{jk}\nabla f(\mathbf{x}_k)] \\
&+ [\frac{|\mathcal{B}|}{n}\sum_{j \neq i^\dagger, j \in \mathcal{N}(i^\dagger)}\left(\sum_{k \in \mathcal{N}(j)\setminus i^\dagger} \mathbf{\Pi}_{jk}\nabla f(\mathbf{x}_k) + \mathbf{\Pi}_{ji^\dagger}\nabla f(\mathbf{x}_{i^\dagger})\right)] \\
&+ [\frac{|\mathcal{B}|}{n}\left(\sum_{k \in \mathcal{N}(i^\dagger)\setminus i^\dagger} \mathbf{\Pi}_{i^\dagger k}\nabla f(\mathbf{x}_k) + \mathbf{\Pi}_{i^\dagger i^\dagger}\nabla f(\mathbf{x}_{i^\dagger})\right)],
\end{aligned} \tag{5}
$$

which indicates $\mathcal{G}(\mathbb{D}) \sim \mu_0$.

$$\mathbb{E}\left[\mathcal{G}(\mathbb{D}')\right] = \left[\frac{|\mathcal{B}|}{n} \sum_{j \neq i^\dagger, j \notin \mathcal{N}(i^\dagger)} G_j\right] + \left[\frac{|\mathcal{B}|}{n} \sum_{j \neq i^\dagger, j \in \mathcal{N}(i^\dagger)} G_j'\right] + \left[\frac{|\mathcal{B}|}{n} G_i'\right]$$

$$= \left[\frac{|\mathcal{B}|}{n} \sum_{j \neq i^\dagger, j \notin \mathcal{N}(i^\dagger)} \sum_{k \in \mathcal{N}(j)} \Pi_{jk} \nabla f(\mathbf{x}_k)\right]$$

$$+ \left[\frac{|\mathcal{B}|}{n} \sum_{j \neq i^\dagger, j \in \mathcal{N}(i^\dagger)} \left(\sum_{k \in \mathcal{N}(j) \setminus i^\dagger} \Pi_{jk} \nabla f(\mathbf{x}_k) + \Pi_{ji} \nabla f\left(\mathbf{x}_{i^\dagger}'\right)\right)\right] \quad (6)$$

$$+ \left[\frac{|\mathcal{B}|}{n} \left(\sum_{k \in \mathcal{N}(i^\dagger) \setminus i^\dagger} \Pi_{i^\dagger k} \nabla f(\mathbf{x}_k) + \Pi_{i^\dagger i^\dagger} \nabla f\left(\mathbf{x}_{i^\dagger}'\right)\right)\right]$$

$$= \mathbb{E}[\mathcal{G}(\mathbb{D})] + \frac{|\mathcal{B}|}{n} \sum_{j=1}^{n} \Pi_{ji^\dagger} \left(f\left(\mathbf{x}_{i^\dagger}'\right) - f(\mathbf{x}_{i^\dagger})\right)$$

$$= \mathbb{E}[\mathcal{G}(\mathbb{D})] + \frac{|\mathcal{B}|}{n} \left\|\Pi_{:,i^\dagger}\right\|_1 \leq \mathbb{E}[\mathcal{G}(\mathbb{D})] + \frac{|\mathcal{B}|}{n} \tau,$$

which indicates $\mathcal{G}(\mathbb{D}') \sim \mu_0 + \frac{|\mathcal{B}|}{n} \mu_\tau$.

In the following, we quantify the divergence between $\mathcal{G}$ and $\mathcal{G}'$ by following the moments accountant paper, where we show that $\mathbb{E}\left[\left(\frac{\mu(z)}{\mu_0(z)}\right)^\lambda\right] \leq \alpha$, and $\mathbb{E}\left[\left(\frac{\mu_0(z)}{\mu(z)}\right)^\lambda\right] \leq \alpha$, for some explicit $\alpha$. To do so, the following is to be bounded for $v_0$ and $v_1$.

$$\mathbb{E}_{z \sim v_0}\left[\left(\frac{v_0(z)}{v_1(z)}\right)^\lambda\right] = \mathbb{E}_{z \sim v_1}\left[\left(\frac{v_1(z)}{v_0(z)}\right)^{\lambda+1}\right] \quad (7)$$

Following [1], the above can be expanded with binomial expansion, which gives

$$\mathbb{E}_{z \sim v_1}\left[\left(\frac{v_1(z)}{v_0(z)}\right)^{\lambda+1}\right] = \sum_{t=0}^{\lambda+1} (\lambda+1) \mathbb{E}_{z \sim v_1}\left[\left(\frac{v_0 - v_1(z)}{v_1(z)}\right)^t\right] \quad (8)$$
$$= 1 + 0 + T_3 + T_4 + \dots$$

Next, we bound $T_3$ by substituting the pairs of $v_0 = \mu_0, v_1 = \mu$ and $v_0 = \mu, v_1 = \mu_0$ in, and upper bound them, respectively.

For $T_3$, with $v_0 = \mu_0, v_1 = \mu$, we have

$$T_3 = \frac{(\lambda+1)\lambda}{2} \mathbb{E}_{z \sim \mu}\left[\left(\frac{\mu_0(z) - \mu(z)}{\mu(z)}\right)^2\right] = \frac{(\lambda+1)\lambda}{2} \mathbb{E}_{z \sim \mu}\left[\left(\frac{q\mu_\tau(z)}{\mu(z)}\right)^2\right]$$

$$= \frac{q^2(\lambda+1)\lambda}{2} \int_{-\infty}^{+\infty} \frac{(\mu_\tau(z))^2}{\mu_0(z) + q\mu_\tau(z)} dz \leq \frac{q^2(\lambda+1)\lambda}{2} \int_{-\infty}^{+\infty} \frac{(\mu_\tau(z))^2}{\mu_0(z)} dz \quad (9)$$

$$= \frac{q^2(\lambda+1)\lambda}{2} \mathbb{E}_{z \sim \mu_0}\left[\left(\frac{\mu_\tau(z)}{\mu_0(z)}\right)^2\right] = \frac{q^2(\lambda+1)\lambda}{2} \exp\left(\frac{\tau^2}{\sigma^2}\right)$$

$$\leq \frac{q^2(\lambda+1)\lambda}{2} \left(\frac{\tau^2}{\sigma^2} + 1\right) \leq \frac{q^2 \tau^2 (\lambda+1)\lambda}{\sigma^2},$$

where in the last inequality, we assume $\frac{\tau^2}{\sigma^2} + 1 \leq 2\frac{\tau^2}{\sigma^2}$, i.e., $\frac{\tau^2}{\sigma^2} \geq 1$. Thus, it requires $\sigma \leq \tau$.

As a result,

$$\alpha_\mathcal{G}(\lambda) \leq \frac{q^2 \tau^2 (\lambda+1)\lambda}{\sigma^2} + O\left(q^3 \lambda^3 / \sigma^3\right). \quad (10)$$

To satisfy $T \frac{q^2 \tau^2 \lambda^2}{\sigma^2} \leq \frac{\lambda \epsilon_{sgd}}{2}$, and $\exp\left(-\frac{\lambda \epsilon_{sgd}}{2}\right) \leq \delta_{sgd}$, we set

$$\epsilon_{sgd} = c_1 q^2 \tau^2 T, \quad (11)$$

$$\sigma = c_2 \frac{q \tau \sqrt{T \log(1/\delta_{sgd})}}{\epsilon_{sgd}}. \quad (12)$$

Given the input DP APPR matrix costs additional $(\epsilon_{pr}, \delta_{pr})$ privacy budget, by using the standard composition theorem of DP, we get the total privacy budget by $G$ is $(\epsilon_{sgd} + \epsilon_{pr}, \delta_{sgd} + \delta_{pr})$. Since $G$ is randomly sampled from the graph dataset $\overline{G}$, we can conclude the proof with the privacy amplification theorem of DP [5, 23]. □

## 7.4 Datasets

We evaluate our method on five graph datasets: Cora-ML [6] which consists of academic research papers from various machine learning conferences and their citation relationships, Microsoft Academic graph [35] which contains scholarly data from various sources and the relationships between them, CS and Physics [35] which are co-authorship graphs, Reddit [20] which is constructed from Reddit posts, where edges represent connections between posts when the same user commented on both. Table 2 shows the statistics of the five datasets.

**Table 2: Dataset statistics**

| Dataset | Cora-ML | MS Academic | CS | Reddit | Physics |
|---------|---------|-------------|-----|--------|---------|
| Classes | 7 | 15 | 15 | 8 | 8 |
| Features | 2,879 | 6,805 | 6,805 | 602 | 8,415 |
| Nodes | 2,995 | 18,333 | 18,333 | 116,713 | 34,493 |
| Edges | 8,416 | 81,894 | 327,576 | 46,233,380 | 495,924 |

## 7.5 Illustration of Privacy Protection

To provide an intuitive illustration of the privacy protection provided by the DP trained models using our methods, we visualize the t-SNE clustering of training nodes' embeddings generated by the private models with varying $\epsilon$ values in Figure 9 for the Cora-ML dataset. We omit the results for other datasets as they display a similar pattern leading to the same conclusion. The color of each node corresponds to the label of the node. We can observe that when the privacy budget is small ($\epsilon = 1$), the model achieves strong privacy protection, thus it becomes hard to distinguish the training nodes belonging to different classes from each other. Meanwhile, when the privacy guarantee becomes weak ($\epsilon$ becomes larger), embeddings of nodes with the same class label are less obfuscated, hence gradually forming a cluster. This observation demonstrates that the privacy budget used in our proposed methods is correlated with the model's ability to generate private node embeddings, and therefore also associated with the privacy protection effectiveness against adversaries utilizing the generated embeddings to carry out privacy attacks [17, 27].

## 7.6 More Results on Effects of Privacy Parameters

**Batch Size in DP-SGD ($B$).** Figure 10 shows the effect of batch size on the model's test accuracy. According to Theorem 2, given the fixed total privacy budget and epochs, the standard deviation of the Gaussian noise is proportional to the square root of the batch size. Therefore, a larger batch size increases the noise for the gradients. On the other hand, it may provide a more accurate update given more nodes and correlations included in the batch. Hence, we observe a relatively flat curve when the batch size is not super small.

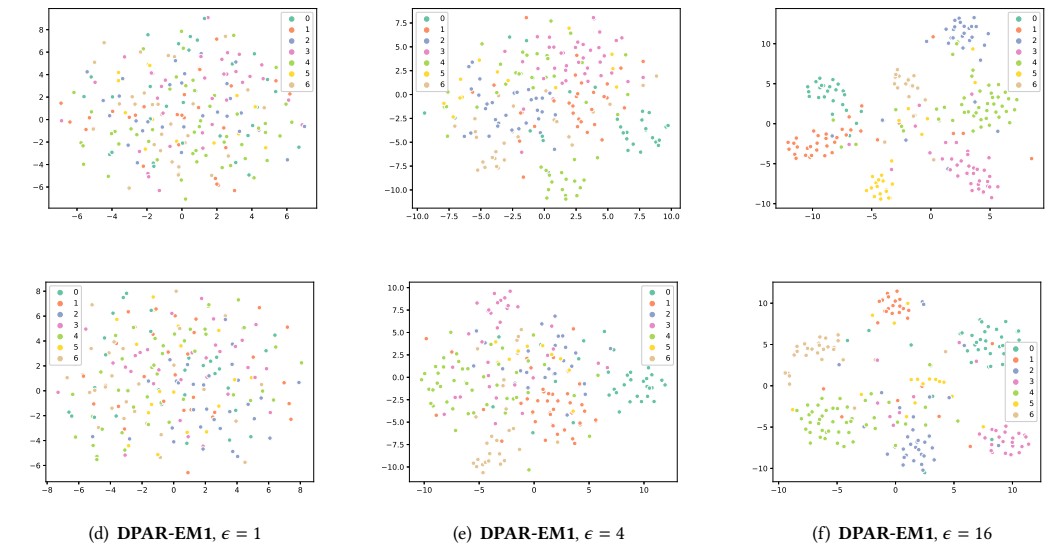

(d) DPAR-EM1, $\epsilon = 1$          (e) DPAR-EM1, $\epsilon = 4$          (f) DPAR-EM1, $\epsilon = 16$

Figure 9: Cora-ML. Clustering of training nodes' embeddings generated by private models with different privacy guarantees $\epsilon$ (fixed $\delta = 2 \times 10^{-3}$) and training methods.

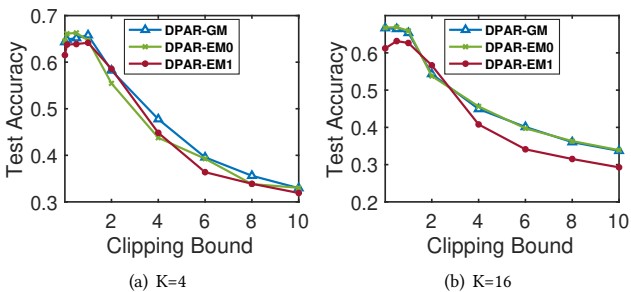

(a) K=4          (b) K=16

Figure 11: Cora-ML. Relationship between clipping bound of DP-SGD and model test accuracy. Fix total privacy budget $(\epsilon, \delta) = (8, 2 \times 10^{-3})$.

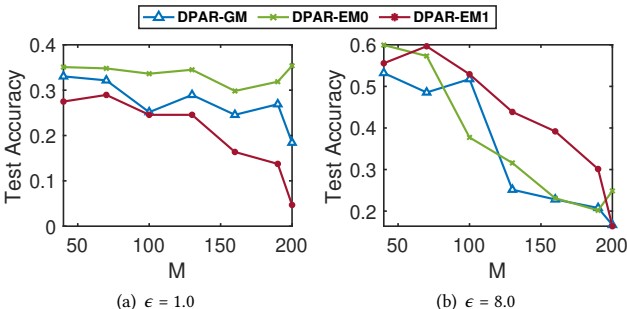

(a) $\epsilon = 1.0$          (b) $\epsilon = 8.0$

Figure 12: Cora-ML: Relationship between the number of nodes M in DP-APPR vector calculation and model test accuracy. K = 2.

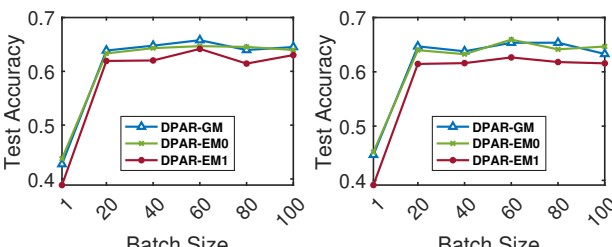

Figure 10: Cora-ML. Batch size vs. model test accuracy. Fix total privacy budget $(\epsilon, \delta) = (8, 2 \times 10^{-3})$. K=4 (left), K=16 (right)

**Clipping Bound in DP-SGD ($C$).** Figure 11 shows the effect of gradient norm clipping bound $C$ in DP-SGD on the model's test accuracy. The clipping bound affects the noise scale added to the gradients (linearly) as well as the optimization direction of model parameters. A large clipping bound may involve too much noise to the gradients, while a small clipping bound may undermine gradients' ability for unbiased estimation. The result verifies this phenomenon. We use $C = 1$ for all datasets in our experiments.

**Number of Nodes in DP-APPR ($M$).** During the DP-APPR algorithm, a subset of M nodes is randomly sampled from the input training graph. Figure 12 illustrates the relationship between M and test accuracy under different total privacy budgets ($\epsilon = 1$ and $\epsilon = 8$, with $\delta = 2 \times 10^{-3}$). As M increases, the privacy budget allocated for calculating each DP-APPR vector decreases. This leads to more noise in each DP-APPR vector, which can adversely affect its utility and result in lower accuracy as observed. However, too small of an $M$ will degrade the performance since it will not contain enough information about the graph structure. In our experiments, we set M = 70 for all datasets.

