# OpenReview forum: "DPAR: Decoupled Graph Neural Networks with Node-Level Differential Privacy"
_ACM.org/TheWebConf/2024/Conference — TheWebConf24 Oral_

### Official Review · Reviewer_q9Ga · 2023-11-22

**Novelty:** 5
**Technical Quality:** 4

**Review:**

Strength:
1. This paper proposed a novel de-coupled DP framework for a better privacy-utility tradeoff.
2. Experiments showed that the proposed method has a better utility-privacy tradeoff.
3. The authors conducted a theoretical analysis of the node DP guarantee.

Weakness:
1. Concerns about the experimental setup: the datasets are splitted into a training set and a test set without a validation set. How to choose the hyperparameters in this paper (i.e., if choosing based on the test performance, would the final reported result be related to the hyperparameter tuning)? What are the hyperparameter spaces for each compared method?
2. Typical GNNs would aggregate information from all neighbors while in this paper, the proposed method will aggregate from top K neighbors (in the default setting, K=2). Could such a limited neighbor make full use of GNN aggregation? What are the K values in Figure 8? It is not clear by directly looking at the figure.
3. In Table 1, the results of models DPARNoDP and GAPNoDP which are the versions of no DP protection are reported. Could you also report the results of other baselines without DP protection so that readers can easily see that the models without DP protection could obtain a similar level of utility performance?

**Questions:**

Refer to weakness

**Reviewer Confidence:**

2: The reviewer is willing to defend the evaluation, but it is likely that the reviewer did not understand parts of the paper

**Scope:**

4: The work is relevant to the Web and to the track, and is of broad interest to the community

---

### Official Review · Reviewer_NLTX · 2023-11-26

**Novelty:** 5
**Technical Quality:** 6

**Review:**

The paper articulates the challenges of achieving node-level differential privacy in GNNs by proposing a novel approach DPAR, which decouples the process of feature aggregation and message passing, preserving structural information and node feature information in a sequential manner. The work contributes to the field by tackling the problem of high node sensitivity and improving the privacy-utility tradeoff. The theoretical analysis and empirical results support the effectiveness of the proposed approach, highlighting its potential impact within practical applications. Refer to Questions for my concerns.

**Questions:**

1.	Is it possible to apply the proposed method under the transductive setting?
2.	How resilient is the proposed method against existing privacy attacks?

**Reviewer Confidence:**

3: The reviewer is confident but not certain that the evaluation is correct

**Scope:**

4: The work is relevant to the Web and to the track, and is of broad interest to the community

---

### Official Review · Reviewer_HJ71 · 2023-11-27

**Novelty:** 6
**Technical Quality:** 6

**Review:**

Quality:
● The paper demonstrates a high level of quality in its approach to addressing the
problem of private learning for Graph Neural Networks (GNNs).
● The proposed method, based on a two-stage framework comprising DP-APPR
and DP-SGD, is well-formulated and systematically presented.
● The mathematical foundations, including the differential privacy mechanisms, are
sound.
● The experiments conducted on real-world graph datasets contribute to the overall
quality, providing empirical evidence of the proposed method's effectiveness.
Clarity:
● The paper is generally well-written. The structure is logical, with a clear
progression from the abstract to the experimental results.
● The mathematical notation is appropriate and contributes to the precision of the
presentation. However, to enhance accessibility, it might be beneficial to include
brief intuitive explanations or summaries at the end of complex proofs.
● Additionally, some sections, particularly in the experimental results, are quite
detailed, and a more concise presentation could improve clarity without
sacrificing content.
Originality:
● The paper presents a novel approach to addressing the privacy challenges
associated with training GNNs. The combination of DP-APPR and DP-SGD,
along with the proposed DP-APPR algorithms using Gaussian and exponential
mechanisms, demonstrates a unique contribution to the field.
Significance:
● The significance of the work is notable, given the increasing importance of
privacy-preserving machine learning, particularly in the context of graph data.
The paper addresses a relevant and challenging problem, offering a solution that
balances privacy and utility for GNNs.
● The empirical results on real-world datasets contribute to the significance,
showcasing the practical viability of the proposed method. The comparison with
state-of-the-art methods and the achieved improvements in test accuracy under
privacy constraints enhance the paper's impact.
Pros:
● Novel Methodology: The combination of DP-APPR and DP-SGD in a two-stage
framework is a novel and effective approach.
● Decoupled Framework: The decoupled framework, allowing independent
protection of graph structure information and node features, is a strength. This
flexibility enables more precise and tunable privacy protection for graph data with
varying characteristics
● Empirical Validation: The experimental results on real-world datasets provide
concrete evidence of the proposed method's effectiveness.
● Clear Presentation: The paper is well-organized, and the mathematical
foundations are presented clearly.
Cons:
● Detail in Experimental Section: Some sections in the experimental results are
overly detailed. A more concise presentation without compromising information
could enhance readability.
● Intuitive Explanations: While the paper is well-structured and clear, including brief
intuitive explanations after complex proofs could improve accessibility for a
broader audience.
● Algorithmic Complexity: The paper could benefit from a more detailed discussion
or analysis of the computational complexity of the proposed algorithm.
Understanding the time and space complexity is crucial for assessing its
scalability to larger datasets.
● Scalability: While the experimental results are promising, the scalability of the
proposed approach to larger graphs or datasets is not extensively discussed.
Most of the experiment figures are provided for the “Vora-ML” dataset which is
the smallest one. Providing insights into how the method performs with varying
graph sizes would enhance the practical applicability of the proposed solution.

**Questions:**

● Algorithmic Complexity: Could you provide a more detailed analysis of the
computational complexity of your proposed algorithm?
● Generalization: How well does your proposed method generalize across different
types of graphs or datasets?
● Scalability: How does the proposed approach scale with increasing graph sizes
or more extensive datasets? Could you discuss any potential challenges or
optimizations for handling larger graphs?

**Reviewer Confidence:**

4: The reviewer is certain that the evaluation is correct and very familiar with the relevant literature

**Scope:**

4: The work is relevant to the Web and to the track, and is of broad interest to the community

---

### Official Review · Reviewer_WtfQ · 2023-11-29

**Novelty:** 5
**Technical Quality:** 5

**Review:**

1) The paper introduces a decoupled approach to Graph Neural Networks (GNNs) for achieving node-level differential privacy (DP)​​. While this approach is novel, its generalizability across different graph structures and types needs further exploration. The effectiveness of decoupling feature projection and message passing could vary significantly across heterogeneous graphs, dynamic graphs, or graphs with high-dimensional features. The paper should consider these varying graph types and structures to evaluate the robustness of their approach. Or the authors need to justify how the proposed methods can be generalize to graphs with different properties to show the robustness.

2) The implementation of differential privacy (DP) in the context of GNNs, particularly the Decoupled GNN with Differentially Private Approximate Personalized PageRank (DPAR) proposed in the paper​​, introduces significant complexity. This complexity could impede the practical application of the method, particularly for users without extensive expertise in DP or GNNs. The paper could benefit from a more detailed discussion on the practical aspects of implementing DPAR, including computational overhead, scalability concerns, and the implications of these factors on large-scale real-world datasets.

3) The scalability and efficiency of the DPAR approach in large-scale graph applications need further investigation. This includes assessing the computational requirements, memory usage, and the feasibility of applying the approach to graphs with millions of nodes and edges. The paper should provide a detailed analysis of these aspects to demonstrate the practicality of DPAR in large-scale applications.

4) The paper introduces a novel approach but does not provide a detailed analysis of the algorithmic complexity of the DPAR method. For practical application, it is crucial to understand the computational cost, especially in terms of time and space complexity. This information is vital for practitioners to gauge the feasibility of deploying this method in real-world scenarios, particularly in cases where computational resources are limited. A thorough complexity analysis would help in comparing the DPAR method with existing GNN approaches in terms of efficiency.

5) While the paper compares the DPAR method with existing approaches, the range of comparisons might be limited. A more extensive set of comparisons with a broader range of state-of-the-art methods [a][b] in both node-level differential privacy and graph neural network architectures would be beneficial. This would include not only comparisons in terms of privacy-utility trade-offs but also in aspects such as training efficiency, scalability to larger graphs, and adaptability to different types of graph data. Such comprehensive comparisons would better position the DPAR method within the current landscape of GNN research and applications.

[a] "ProGAP: Progressive Graph Neural Networks with Differential Privacy Guarantees" WSDM 2024
[b] "Local Differential Privacy in Graph Neural Networks: a Reconstruction Approach" arXiv:2309.08569

**Questions:**

1) How generalizable is the decoupled approach to Graph Neural Networks (GNNs) in the context of varying graph structures and types, including heterogeneous, dynamic, or high-dimensional feature graphs?

2) Can you provide a more detailed discussion on the practical aspects of implementing the Decoupled GNN with Differentially Private Approximate Personalized PageRank (DPAR), particularly concerning computational overhead, scalability, and its implications for large-scale real-world datasets?

3) How scalable and efficient is the DPAR approach in handling large-scale graph applications, especially regarding computational requirements, memory usage, and applicability to graphs with millions of nodes and edges?

4) Could you elaborate on the algorithmic complexity of the DPAR method in terms of time and space, to better understand its feasibility and efficiency compared to existing GNN approaches?

5) Is it possible to extend the range of comparisons for the DPAR method to include a broader spectrum of state-of-the-art methods in node-level differential privacy and graph neural network architectures, focusing on aspects like privacy-utility trade-offs, training efficiency, scalability, and adaptability to different graph data types?

**Reviewer Confidence:**

3: The reviewer is confident but not certain that the evaluation is correct

**Scope:**

3: The work is somewhat relevant to the Web and to the track, and is of narrow interest to a sub-community

---

### Official Review · Reviewer_a9rx · 2023-11-30

**Novelty:** 7
**Technical Quality:** 7

**Review:**

The authors present a a new approach for differentially-private training of a graph neural network. Different from differential privacy in non-graph neural networks, the problem becomes much more challenging because gradients depend not only on the current node, but also on neighbors, which could be very many in number. Existing graph neural network differential privacy methods limit the hops or neighbors, which yields poor accuracy of the model. This paper does Approximate Personalized PageRank (and a differentially-private version of it at that) to overcome the limitation.

I really liked this paper. It was well-written and explained the context of the problem, the challenges, and the solution really well. The proposed algorithm is well done and requires ingenuity beyond just gluing a couple of things together. The theoretical analysis is great and the empirical results are comprehensive and compelling.

**Questions:**

- is the acronym pronounced "dapper"? if not, maybe you should suggest to readers that it should be.

**Ethics Review Description:**

-

**Reviewer Confidence:**

3: The reviewer is confident but not certain that the evaluation is correct

**Scope:**

4: The work is relevant to the Web and to the track, and is of broad interest to the community

---

### Decision · Program_Chairs · 2024-01-22

**Decision:**

Accept (Oral)

**Comment:**

Our decision is to accept. Please see the AC's review below and improve the work considering that and the reviewers' feedback for cemera-ready submission.

"This paper presents a new approach for node-level differential privacy in Graph Neural Networks (GNNs), focusing on the challenge posed by the dependence of the gradient on not just a given node but the node's neighborhood (which may be very large). To circumvent this issue, the authors propose a ""decoupled"" approach to differential privacy training based on PageRank, known as DPAR, which is novel and interpretable. The authors complement the description of their method with a concrete theoretical analysis and comprehensive empirical investigation. Overall, I found this paper compelling both in terms of the theoretical and empirical backing of DPAR as a novel and effective approach to addressing privacy concerns in GNNs.

 While the paper is generally well-written, it is not the most accessible, and could benefit from more intuitive explanations and proof sketches (to help a broader audience understand the complexity of the proofs in the paper). There are many details in the experimental section which is great for completeness, but could potentially be moved to an appendix, in-line with the suggestions from R3. The authors might also consider providing additional clarification to how DPAR might generalize to various types of graphs and datasets (e.g., heterogenous, dynamic, or high-dimensional feature graphs), as well as discuss potential scalability issues with large graphs or data.

 Due to the complexity of DPAR, it may be worth discussing implementation concerns and computational overhead relative to more baseline GNN methods. It may also be worth furthering the theoretical analysis by exploring memory usage and algorithmic complexity of the method, as R2 and R3 point out. I also find R4's suggestion about exploring the resilience of the proposed method against existing privacy attacks, and its applicability to the transductive setting, to be an interesting future extension."